# Associations of Dietary Patterns during Pregnancy with Gestational Hypertension: The “Born in Shenyang” Cohort Study

**DOI:** 10.3390/nu14204342

**Published:** 2022-10-17

**Authors:** Jiajin Hu, Lin Li, Ningyu Wan, Borui Liu, Yilin Liu, Yanan Ma, Chong Qiao, Caixia Liu, Deliang Wen

**Affiliations:** 1Health Sciences Institute, China Medical University, Shenyang 110122, China; 2Research Center of China Medical University Birth Cohort, China Medical University, Shenyang 110122, China; 3Division of Chronic Disease Research Across the Lifecourse, Department of Population Medicine, Harvard Medical School, Boston, MA 02215, USA; 4Department of Developmental Pediatrics, Shengjing Hospital of China Medical University, China Medical University, Shenyang 110004, China; 5Department of Obstetrics and Gynecology, Shengjing Hospital of China Medical University, China Medical University, Shenyang 110004, China; 6Department of Epidemiology and Health Statistics, School of Public Health, China Medical University, Shenyang 110122, China

**Keywords:** pregnancy, pre-pregnancy BMI, maternal nutrition, dietary patterns, gestational hypertension

## Abstract

The literature on maternal dietary patterns and gestational hypertension (GH) risk is largely ambiguous. We investigated the associations of maternal dietary patterns with GH risk among 1092 pregnant women in a Chinese pre-birth cohort. We used both three-day food diaries (TFD) and food frequency questionnaires (FFQ) to assess the diets of pregnant women. Principal components analysis with varimax rotation was used to identify dietary patterns from the TFD and FFQ, respectively. In total, 14.5% of the participants were diagnosed with GH. Maternal adherence to a “Wheaten food–coarse cereals pattern (TFD)” was associated with a lower risk of GH (quartile 3 [Q3] vs. Q1, odds ratio [OR] = 0.53, 95%CI: 0.31, 0.90). Maternal adherence to a “Sweet food–seafood pattern (TFD)” was associated with lower systolic blood pressure (Q4 vs. Q1, β = −2.57, 95%CI: −4.19, −0.96), and mean arterial pressure (Q4 vs. Q1, β = −1.54, 95%CI: −2.70, −0.38). The protective associations of the “Sweet food-seafood (TFD)” and “Fish–seafood pattern (FFQ)” with the risk of GH were more pronounced among women who were overweight/obese before pregnancy (*p* for interaction < 0.05 for all). The findings may help to develop interventions and better identify target populations for hypertension prevention during pregnancy.

## 1. Introduction

Gestational hypertension (GH), a common gestational cardiovascular disease, has short-term and long-term influences on both mothers (e.g., placental abruption, premature delivery) and offspring (e.g., neonatal asphyxia), and has become an important cause of maternal and neonatal mortality [1,2,3,4]. The prevalence of GH varies from 1.8% to 4.4% worldwide, and has been reported to be higher among pregnant women of disadvantaged social-economic status, such as women with lower educational attainment [5]. This could be due to lifestyle differences among populations with different social-economic statuses; therefore, the investigation of lifestyle factors in relation to maternal blood pressure may contribute to the development of strategies to prevent hypertensive disorders during pregnancy. 

Maternal dietary intake plays an important role in the development of cardiovascular and metabolic diseases during pregnancy, such as GH [6] and gestational diabetes mellitus (GDM) [7]. Existing studies have shown that the increased intake of single nutrients such as *n*-3 PUFAs [8] and zinc [9], or specific foods such as seafood [10], during pregnancy were associated with a lower risk of GH and pre-eclampsia. However, a single nutrient may not well represent the complicated interactions among food items and nutrients. Instead of looking at single nutrient intake, maternal dietary patterns better reflect women’s overall food consumption during pregnancy and may be more relevant and useful for predicting pregnancy complications, such as GH. Previous studies that have examined the association between maternal dietary patterns and the risk of GH were mainly based on Western populations and have reported inconsistent results [6,11]. For example, several studies have reported that maternal adherence to healthy dietary patterns such as the Mediterranean-style dietary pattern [12,13] or the Dietary Approaches to Stop Hypertension (DASH) Diet [14] is associated with a lower risk of GH; however, in the Generation R study in the Netherlands, the Mediterranean diet was not associated with a lower risk of GH or pre-eclampsia [15]. In the US cohort Project Viva, adherence of pregnant women to high-quality dietary patterns was also not observed to be associated with a lower risk of pre-eclampsia [16]. Only a few studies have examined the association between GH and dietary patterns among the Chinese population [12], which has different dietary culture and food consumption patterns from Western populations. 

In addition, previous studies have reported that the association between maternal diet and pregnancy complications, such as GDM [17] and excessive gestational weight gain [18], may be stratified by women’s pre-pregnancy body mass index (BMI). Possible mechanisms could be that the pre-pregnancy weight status represents the overall nutrient and metabolism background, and thus, may modify the influence of prenatal diets on women’s metabolic outcomes. To our knowledge, no previous study has examined how pre-pregnancy weight status influences the association between GH and maternal dietary patterns, which may help to identify target populations for GH prevention by prenatal dietary interventions.

Previous studies mostly used a single dietary assessment tool to measure gestational diet consumption, e.g., food diaries or food frequency questionnaires (FFQ). However, both food diaries and FFQs carry limitations in assessing food consumption. Food diaries have the advantage of capturing an accurate depiction of food consumption, but fail to address long-term dietary habits. While FFQs provide relatively long-term diet consumption information, they are less capable of accurately capturing food intake, and may overestimate food intake. The joint use of the two tools will give a more comprehensive picture of the diets of pregnant women.

In the present study, we hypothesized that dietary patterns during pregnancy are associated with maternal blood pressure and GH risk. We used both FFQ and three-day food diaries (TFD) to derive dietary patterns and examine the association between prenatal dietary patterns and women’s GH risk in the Born in Shenyang Cohort Study (BISCS), a prospective cohort study in northeast China. 

## 2. Materials and Methods

### 2.1. Study Population

The cohort was conducted based on the national maternal and child health care system in China and the study design of BISCS has been described elsewhere [19]. In brief, we enrolled pregnant women with single pregnancies at 54 community health care centers and hospitals that provide prenatal care services in Shenyang, China, from April to September 2017. To obtain the maximum sample size with better sample representation, we recruited pregnant women between 21 and 24 weeks of gestation, because most pregnant women took the first prenatal care in the second trimester and finished registration in the system before 20 weeks of gestation. Among the 2068 eligible women, 1338 agreed to participate in the project and 1296 pregnant women had singleton live births. In the primary analysis, in which we aimed to assess the association of dietary patterns with mean blood pressure during pregnancy, we excluded women with elevated clinical blood pressure (BP) measurements less than 3 times throughout pregnancy (*n* = 17), and women with incomplete dietary assessments during pregnancy or implausible dietary reporting (<500 or >3500 kcal/day) (*n* = 187), leaving 1092 participants in the analysis. In the secondary analysis, we further excluded women who had elevated clinically measured blood pressure values (systolic blood pressure [SBP] > 140 mm Hg or diastolic blood pressure [DBP] > 90 mm Hg) before 20 weeks of gestation (*n* = 10) and women who were diagnosed with chronic hypertension before pregnancy (*n* = 3), in order to determine the association between maternal dietary patterns and the risk of GH. We compared the characteristics of women who were included in the analysis (*n* = 1092) with women who were excluded (*n* = 204), and observed that participants excluded were of older age, less likely to be of Chinese Han ethnicity, smokers, had lower physical activity levels and higher calorie intakes (Appendix A). All participants signed a written informed consent form, and the Ethics Committee of China Medical University approved the study. 

### 2.2. Exposures

We assessed maternal diets during pregnancy using both three-day food diaries (TFD) and food frequency questionnaires (FFQ). Clinical researchers trained the participants to complete the TFD at the enrollment visit (mean gestation age ± SD: 22 ± 1.2 weeks), and collected the TFDs from pregnant women at the oral glucose tolerance test (24 ± 1.2 weeks). Participants were asked to record all of their food and drink consumption over two consecutive working days and one weekend day, including ingredients and portion sizes. Visual aid manuals, which included photos of 200 different types of food, were provided to participants to help them identify the portion size of food. We categorized the food items from the TFDs into 21 non-overlapping food groups (Appendix A), averaged daily food consumption over three days, and calculated daily energy and nutrient intake based on the Chinese Food Composition database [20]. We also used a semiquantitative FFQ to assess women’s food consumption frequency during pregnancy at the enrollment visit (24 ± 1.2 weeks). Pregnant women were asked to report their food consumption frequency over the last two months. The FFQ contained 25 food items with nine consumption frequency categories ranging from “never” to “more than or equal to 3 times per day”. We further grouped the 25 FFQ food categories into the 21 TFD food groups according to similar nutritional content (e.g., combining pork, beef, and lamb into one red meat group). We conducted a second FFQ (FFQ2) at a mean of 33 weeks of gestation among a subset of 401 pregnant women who were diagnosed with non-GDM to examine the reproducibility of the FFQ. We chose non-GDM women because pregnant women may change their dietary habits after being diagnosed with GDM. The Spearman correlation coefficients of the consumption frequencies (per day) ranged from 0.80 (vegetables) to 0.43 (tubers) between FFQ1 (24 weeks of gestation) and FFQ2 (33 weeks of gestation), which indicated a relatively high reproducibility. We also assessed the validity of the FFQ1 with the TFD. The Spearman correlation coefficients between the two assessment tools ranged from 0.60 (vegetables) to 0.25 (whole grains), which indicated a moderate-to-high level of validity of the tools.

### 2.3. Outcomes

Clinical researchers measured the pregnant women’s blood pressure during each antenatal care visit (the number of measurements ranged from 3 to 13, with a median of 7) and three measurements were obtained during each antenatal care visit. All participants took a 5-min break before measurement. The SBP and DBP were calculated as the average of the three measurements. Mean arterial pressure (MAP) was calculated using 1/3 SBP plus 2/3 DBP. We identified pregnant women as having chronic hypertension if they were diagnosed before pregnancy or had two elevated clinical blood pressure measurements (SBP > 140 mmHg or DBP > 90 mmHg) before 20 weeks of gestation [21]. We defined GH women as those who did not have chronic hypertension but had documented elevated SBP (>140 mmHg) or DBP (>90 mmHg) at least two times (at least 4 h apart) after 20 weeks of gestation [22].

### 2.4. Covariates

A standard questionnaire was used at the enrollment visit to assess the participants’ demographics and social-economic and behavioral information, including the women’s age, race, educational attainment, household income, parity, pre-pregnancy weight, and smoking status. Physical activity during pregnancy was assessed using the Chinese version of the Pregnancy Physical Activity Questionnaire [23]. Clinical researchers measured pregnant women’s height at the enrollment visit using the employed calibrated rangefinders (Seca 217; Seca Corporation, Hamburg, Germany). The women self-reported their pre-pregnancy body weight at the enrollment visit. We calculated the pregnant women’s pre-pregnancy body mass index (BMI) by dividing pre-pregnancy weight (kilograms) by the square of the measured height (meters) and divided maternal pre-pregnancy BMI into three groups (<18.5, 18.5–<24.0, ≥24.0 kg/m^2^) according to the World Health Organization reference standard for Asian populations [24]. We categorized participants’ age into four groups (<25, 25–29, 30–34, ≥35 years), race into two groups (Chinese Han, others), educational attainment into two groups (high school or below, college or above), household income per year into two groups (<¥50,000 and ≥¥50,000), parity into two groups (0 and ≥1), smoking status during pregnancy into two groups (yes and no), and physical activity level into three groups (<100, 100–200, >200 metabolic equivalent [MET]-hour/week). The participants’ daily calorie intake was divided into two groups (<2100 kcal/day or ≥2100 kcal/day) according to the recommendation of Dietary Guidelines for Chinese residents [25]. The women’s history of hypertension (including history of GH) and history of diabetes were obtained from medical records and were categorized into two groups, respectively (yes and no). GDM was tested using a 75 g, 2-h oral glucose tolerance test during 24–28 weeks of gestation and diagnosed according to the International Association of Diabetes Pregnancy Study Group standard: fasting plasma glucose ≥ 5.1 mmol/L, 1-h plasma glucose ≥ 10.0 mmol/L, or 2-h plasma glucose ≥ 8.5 mmol/L [26].

### 2.5. Statistical Analyses

Principal components analysis with varimax rotation was used to identify dietary patterns from the TFD and FFQ, respectively. We standardized the daily food intake from the TFD and the food intake frequency from the FFQ into z-scores, and used the standardized values of food groups to identify dietary patterns. We identified the distinct dietary patterns according to the eigenvalue (>1), the factor interpretability of varimax rotation, and a scree plot (Appendix A). For the FFQ, we calculated the dietary pattern scores by summing the standardized food consumption frequencies’ weighted corresponding factor loadings. For the TFD, we calculated the dietary pattern scores by summing the standardized food intakes’ weighted corresponding factor loadings. A food item with an absolute value of factor loading larger than 0.20 was defined as the main contributor to the dietary pattern [27]. We categorized dietary pattern scores into quartiles for further analysis.

T-tests or ANOVA tests were used to compare dietary pattern scores across women’s social demographic characteristics. Multivariable linear regression models were used to estimate associations between dietary pattern scores (in quartiles) and women’s mean SBP, DBP, and MAP throughout pregnancy. Logistic regression models were used to estimate the odds ratio (OR) and the 95% confidence interval (CI) for GH in relation to maternal dietary pattern quartiles. We performed crude and adjusted analyses based on the following models: Model 1, the crude model; Model 2, adjusted for other dietary patterns from the corresponding dietary assessment tool (TFD or FFQ); and Model 3, further adjusted for women’s age, parity, family income, education attainment, race, smoking status, total calorie intake per day and physical activity status per week, pre-pregnancy BMI, history of hypertension, and history of diabetes based on Model 2. In the sensitivity analysis, we further adjusted the gestational weight gain and GDM status of pregnant women, respectively, to examine whether the association could be explained by weight gain or blood-glucose status during pregnancy. To assess the linear trend between dietary patterns and pregnant women’s blood pressure, we took the median score of each dietary pattern quartile as a continuous variable, and reported the *p*_-for-trend_ in multivariable models. Previous studies have reported that associations between maternal diet and complications of pregnancy may be modified by women’s pre-pregnancy weight status [17,18]. Therefore, we further examined the potential effect modification of maternal pre-pregnancy BMI (overweight/obese [OwOb] [BMI < 24 kg/m^2^] vs. non-OwOb [BMI ≥ 24 kg/m^2^]) by including multiplicative interaction terms in the models. All analyses were conducted using Stata S.E. version 16 (Stata Corp, College Station, TX, USA).

## 3. Results

### 3.1. Characteristics of Participants

Among the 1092 pregnant women, 158 (14.5%) were diagnosed as having GH. The maternal mean (±SD) DBP, SBP, and MAP throughout pregnancy were 70.6 (±6.4), 112.4 (±9.9), and 84.0 (±7.1) mmHg, respectively. The participants’ social-economic and physiological characteristics were presented in Table 1. 

### 3.2. Dietary Patterns

We identified four dietary patterns from the TFD and three dietary patterns from the FFQ, respectively (Appendix A). The four dietary patterns derived from the TFD accounted for 26.7% of the total variation and were named the “Traditional pattern (TFD)” (higher intakes of tubers, vegetables, fruits, red meat, coarse cereals, rice, and nuts), “Wheaten food-coarse cereals pattern (TFD)” (higher intakes of wheat flour and products, coarse cereals, beans, bean products, and low intakes of egg and rice), “Sweet food–seafood pattern (TFD)” (higher intakes of pastries and candies, sweet beverages, shrimps, crabs, mussels, and fruits), and “Fried food–protein-rich pattern (TFD)” (higher intakes of fried foods, beans, and dairy products). The three dietary patterns derived from the FFQ accounted for 35.9% of the total variation and were named the “Fish-seafood pattern (FFQ)” (higher intakes of shrimps, crabs, mussels, marine fish, freshwater fish, organ meat, seaweed, and poultry), “Protein-rich pattern (FFQ)” (higher intakes of dairy products, milk, eggs, beans, nuts, pastries, and candies), and “Vegetable–fruit–rice pattern (FFQ)” (higher intakes of vegetables, fruits, rice, and nuts). The maternal dietary pattern scores according to the characteristic categories of pregnant women are presented in Table 2. 

### 3.3. Dietary Patterns in Relation to Blood Pressures during Pregnancy and Risk of GH

Table 3 presents the associations of maternal dietary pattern scores with average blood pressure throughout pregnancy. After adjusting for confounders, we observed that pregnant women who adhered to the “Sweet food–seafood pattern (TFD)” had lower SBP (Q4 vs. Q1, β = −2.57, 95%CI: −4.19, −0.96, *p* for trend = 0.005), and MAP (Q4 vs. Q1, β = −1.54, 95%CI: −2.70, −0.38, *p* for trend = 0.009). Maternal adherence to the “Fish–seafood pattern (FFQ)” was associated with lower SBP (Q2 vs. Q1, β = −2.17, 95%CI: −3.84, −0.50). In the sensitivity analysis, further adjustment for gestational weight gain or GDM did not change the results in general (Appendix A). 

We also observed that women in higher “Wheaten food–coarse cereals pattern (TFD)” quartiles were at lower odds of GH (Q3 vs. Q1, odds ratio [OR] = 0.53, 95%CI: 0.31, 0.90). No significant associations were observed for other dietary patterns in relation to the risk of GH (Table 4).

### 3.4. Dietary Patterns in Relation to GH Stratified by Women’s Pre-Pregnancy Weight Status

In the subgroup analysis according to women’s pre-pregnancy weight status, we observed that the protective associations of the “Sweet food-seafood (TFD)” and “Fish–seafood pattern (FFQ)” with GH were more pronounced among women who were OwOb before pregnancy (*p* _for interaction_ < 0.05 for all) (Figure 1). Compared with women in the lowest quartile (Q1) of the “Sweet food-seafood (TFD)” score, the adjusted ORs for the highest quartile (Q4) were 0.30 (95%CI: 0.16, 1.00) among pre-pregnancy OwOb women and 1.36 (95%CI: 0.70, 2.66) among pre-pregnancy non-OwOb women. For the “Fish–seafood pattern (FFQ)”, the adjusted ORs for Q3 were 0.40 (95%CI: 0.16, 0.99) among pre-pregnancy OwOb women and 2.12 (95%CI: 0.98, 4.59) among pre-pregnancy non-OwOb women. 

## 4. Discussion

In this prospective pre-birth cohort in China, we observed that maternal adherence to a “Wheaten food–coarse cereals pattern (TFD)” was associated with a lower risk of gestational hypertension. Maternal adherence to a “Sweet food–seafood pattern (TFD)” was associated with lower systolic blood pressure, and mean arterial pressure. Adherence to a “Fish–seafood pattern (FFQ)” was associated with lower systolic blood pressure. The protective association of the “Sweet food–seafood pattern (TFD)”, and “Fish–seafood pattern (FFQ)” with the risk of gestational hypertension was more pronounced among women who were overweight/obese before pregnancy.

In our study, eating a “Wheaten food–coarse cereals pattern (TFD)” during pregnancy, which is characterized by a higher intake of wheat flour and products, coarse cereals, and a lower intake of rice, was associated with a lower risk of GH. Previous studies have shown an association between a higher coarse-grain intake and a decreased hypertension risk among adults, which is consistent with our results [28,29]. In the Chinese Kadoorie Biobank study [28], when compared with participants who reported never consuming coarse grain, participants with a daily coarse-grain intake had decreased DBP and SBP and were at a lower risk of hypertension. In the China Health and Nutrition Survey [29], adherence to a coarse cereals dietary pattern was also reported to be associated with decreased risk of hypertension among the general population. A meta-analysis suggested that the consumption of higher levels of whole grains, nuts, and legumes before and during pregnancy, and lower consumption of meat and fine grains, was associated with a reduced risk of GH. The potential mechanisms could be that coarse grains, such as millet, sorghum, and corn, have relatively high levels of dietary fiber and potassium and low glycemic index (GI) [30], giving a protective profile against the development of GH. A high-fiber diet may also inhibit cholesterol absorption, helping to reduce blood pressure, and may prevent GH by altering the gut microbiota, which has been reported to be associated with the renin–angiotensin–aldosterone system, the main physiological pathway of blood pressure homeostasis [31]. A recent cluster-randomized trial [32] reported that a higher dietary potassium supplementation may reduce major cardiovascular events, including high blood pressure. Dietary potassium has been shown to exert a powerful, dose-dependent inhibitory effect on sodium sensitivity [33], which is the main dietary risk factor for hypertension development. In addition, dietary patterns with higher coarse grains and lower refined grains, such as rice, were also characterized by lower GI. The low-GI diet may alter women’s cardiovascular risk toward a protective profile by reducing the risk of excessive weight gain during pregnancy [34]; however, in our study, further adjustment for gestational weight gain did not change the inverse association between the “Wheaten food–coarse cereals pattern (TFD)” score and women’s mean blood pressure. 

In our study, maternal adherence to dietary patterns with a higher intake of seafood, such as the “Sweet food–seafood pattern (TFD)” and the “Fish–seafood pattern (FFQ)”, were consistently linked with lower blood pressure during pregnancy, even though the “Sweet food–seafood pattern (TFD)” was also characterized by high consumption of sweet foods, which are known to be a risk factor for cardiovascular diseases [35]. Our results are consistent with previous studies [8,10,11,36] that indicated that a “Seafood diet”, characterized by high fish and vegetable consumption, was related to a lower risk of GH. In another study, a Nordic diet with high seafood intake was also reported to have a protective association with preeclampsia [37]. Fish and seafoods are low in fat and rich in high-quality protein, which may reduce blood lipids and delay atherosclerosis. Fish and seafoods are also rich in nutrients including zinc, magnesium, and calcium, which have a positive effect on hypertension prevention [38,39]. Previous studies have indicated that adequate zinc intake has a protective effect against the occurrence of GH [9]. Meta-analyses have also suggested that high dietary calcium [8,39,40] and magnesium intakes [8] are related to a lower occurrence of GH and a decreased risk of preeclampsia [40,41,42]. The possible mechanisms could be that, in the process of the development of hypertension, the concentration of free Ca^2+^ is closely related to the dysfunction of vascular smooth-muscle contraction and has regulatory roles in the contraction process and the contractile ability of smooth-muscle cells. The influence of magnesium [39] on blood pressure is that Ca-Mg-ATP constitutes a calcium pump. Magnesium (Mg) acts as a natural calcium channel blocker, which can increase nitric oxide levels and reduce endothelial dysfunction, regulate the calcium concentration of vascular smooth muscle cells, and prevent vascular calcification [41,42]. Magnesium can also inhibit the release of acetylcholine and adrenaline, control muscle excitement and blood vessel spasm, and adjust blood vessel tension to stabilize blood pressure. However, some studies have suggested that seafood consumption is not associated with pregnancy complications such as GH [43], and further studies are warranted.

In the present study, we observed evidence that the association between maternal dietary patterns and the development of GH is stratified by maternal pre-pregnancy weight status. Previous studies have indicated that the association between maternal prenatal dietary patterns and the risk of GDM [17] or gestational weight gain [18] is stratified by women’s BMI levels. To our knowledge, ours is the first study to report the effect of the interaction between maternal dietary patterns and pre-pregnancy weight status on maternal GH risk. The protective association of the “Sweet food–seafood pattern (TFD)” and “Fish–seafood pattern (FFQ)” with GH risk was generally more pronounced among women who were OwOb before pregnancy. Though the mechanisms of the interaction remain unclear, we speculate that women OwOb before pregnancy are more likely to be in a critical physiological state of blood-pressure disorders, and thus may be more sensitive to behavioral factors, such as prenatal diet. The findings of our study indicate that adherence to healthy dietary patterns during pregnancy may benefit women who are already OwOb from developing GH. 

The strengths of the present study include the population-based prospective design, the repeated research-standard measures of gestational blood pressure, and the joint use of the TFD and the FFQ, which provided both an accurate assessment of short-term food intake (TFD) and the habitual diet throughout pregnancy (FFQ). Our study has several limitations. First, limited by the missing information on women’s proteinuria in the medical record, we did not assess the association between maternal dietary patterns and the risk of pre-eclampsia, which is a better indicator for adverse maternal and fetal outcomes. We will address this association in a further study. Secondly, the self-reported pre-pregnancy weight status may cause recall bias in the present analysis. However, several previous studies have indicated a high correlation between the self-reported and measured pre-pregnancy weight statuses [44,45]. Third, the study was based on an urban setting, with participants of relatively high social-economic status, and thus may not be generalizable to other populations. Fourth, we did not measure women’s postpartum blood pressure, which may help to distinguish GH from chronic hypertension. Fifth, the TFD and the FFQ were both conducted mid-pregnancy and therefore may not adequately capture the women’s diet during late pregnancy. However, previous studies have reported that dietary patterns are likely to remain stable throughout pregnancy [46,47]. Sixth, the total variation explained by the dietary patterns identified was relatively small (26.8% for the four TFD patterns and 35.6% for the three FFQ patterns), which is common in studies of dietary patterns among the Chinese population. One possible reason could be that the complexities of Chinese dietary and other food consumption combinations were not identified as distinct dietary patterns. Finally, there may be residual confounding factors, such as the genetic risk of hypertension, which we did not take into account in this analysis.

## 5. Conclusions

In summary, we observed several maternal dietary patterns associated with lower prenatal blood pressure and GH risk. Maternal adherence to dietary patterns with the high consumption of fish and seafoods (e.g., Sweet food–seafood pattern and Fish–seafood pattern) was associated with a lower risk of prenatal blood pressure. Maternal adherence to the “Wheaten food–coarse cereals pattern (TFD)” may be of benefit to women who are at high risk of GH. The protective association of the “Sweet food–seafood pattern (TFD)”, and “Fish–seafood pattern (FFQ)” with the risk of GH were more pronounced among women who were OwOb before pregnancy, which indicated OwOb women may derive more benefit from high seafood consumption. Our findings may help in developing preventive strategies to reduce GH and identify the target population for dietary intervention.

## Figures and Tables

**Figure 1 nutrients-14-04342-f001:**
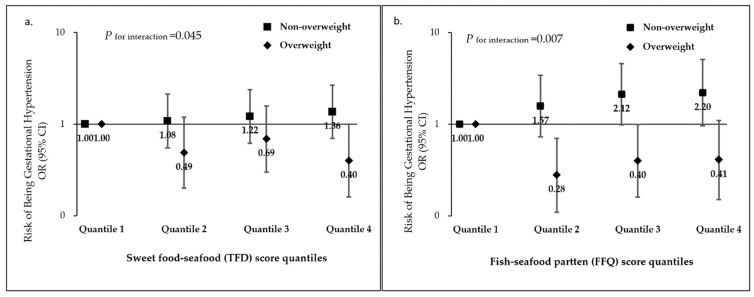
Associations of maternal (**a**) Sweet food-seafood (TFD), (**b**) Fish–seafood pattern (FFQ) scores during pregnancy, in quartiles, with risk of gestational hypertension, stratified by pre-pregnancy weight (overweight vs. non-overweight). OR: odds ratio, CI: confidence interval, TFD: three-day food dairies, FFQ: food frequency questionnaire, OwOb: overweight/obese.

**Table 1 nutrients-14-04342-t001:** Characteristics of pregnant women in the “Born in Shenyang” cohort study (*n* = 1092).

Characteristics	Mean ± SD or *n* (%)
Age at enrollment (Years)	
<25	64 (5.9)
25–29	473 (43.3)
30–35	385 (35.2)
>35	170 (15.6)
Ethnicity	
Chinese Han	911 (83.4)
Others	181 (16.6)
Educational attainment	
High school or below	262 (24.0)
College or above	830 (76.0)
Household income per year, CNY	
<50,000	585 (53.6)
≥50,000	507 (46.4)
Parity	
Primiparous	847 (77.6)
Multiparous	245 (22.4)
Smoking status	
Yes	7 (0.6)
No	1085 (99.4)
Pre-pregnancy BMI category	
<18.5, kg/m^2^	137 (12.6)
18.5-<24.0, kg/m^2^	665 (60.9)
≥24.0, kg/m^2^	290 (26.5)
Physical Activity	
<100 MET-hour/week	282 (25.8)
100 to <200 MET-hour/week	594 (54.4)
≥200 MET-hour/week	216 (19.8)
Energy intake	
<2100 Kcal/d	683 (62.5)
≥2100 Kcal/d	409 (37.5)
History of hypertension	
Yes	3 (0.3)
No	1089 (99.7)
History of diabetes	
Yes	2 (0.2)
No	1090 (99.8)
Gestational hypertension	
Yes	158 (14.5)
No	934 (85.5)
Gestational diabetes mellitus	
Yes	213 (22.3)
No	743 (77.7)
Diastolic blood pressure (mmHg)	70.0 ± 6.4
Systolic blood pressure (mmHg)	112.4 ± 10.0
Mean arterial pressure (mmHg)	84.0 ± 7.1

SD: standard deviation, CNY: Chinese Yuan, BMI: body mass index, MET: metabolic equivalent.

**Table 2 nutrients-14-04342-t002:** Dietary pattern scores according to participants’ characteristics among 1089 participants in the “Born in Shenyang” cohort.

Characteristics	Dietary Pattern Scores, Mean (SD)						
	Three-Day Food Diaries				Food Frequency Questionnaires		
	Traditional (TFD)	Wheaten Food-Coarse Cereals (TFD)	Sweet Food-Seafood (TFD)	Fried Food-Protein Rich (TFD)	Fish-Seafood (FFQ)	Protein-Rich (FFQ)	Vegetable-Fruit-Rice (FFQ)
Age at enrollment, years							
<25	0.25 (1.61)	−0.29 (1.30)	0.10 (1.51)	0.17 (1.75)	−0.36 (2.84)	−0.46 (2.29)	−0.63 (1.83)
25–29	0.06 (1.78)	−0.09 (1.22)	0.05 (1.40)	0.03 (1.25)	0.09 (4.04)	0.70 (2.70)	0.22 (2.02)
30–34	−0.08 (1.65)	0.14 (1.50)	−0.04 (1.14)	0.03 (1.24)	−0.04 (3.50)	−0.14 (2.16)	−0.12 (1.91)
≥35	−0.10 (1.82)	0.03 (1.28)	−0.09 (1.48)	−0.21 (1.17)	−0.03 (4.14)	−0.25 (2.54)	−0.08 (2.03)
*p*	0.344	0.022	0.516	0.099	0.839	0.012	0.003
Ethnicity							
Han	−0.01 (1.72)	0.03 (1.32)	0.00 (1.34)	0.01 (1.29)	0.04 (3.90)	−0.01 (2.47)	−0.03 (1.97)
Minority	0.05 (1.72)	−0.17 (1.45)	0.00 (1.31)	−0.04 (1.17)	−0.18 (3.35)	0.07 (2.53)	0.16 (2.03)
*p*	0.696	0.070	0.961	0.653	0.489	0.668	0.235
Educational attainment							
High school or below	0.23 (1.72)	−0.12 (1.20)	−0.16 (1.10)	−0.12 (1.19)	−0.38 (2.68)	−0.24 (2.50)	−0.21 (2.05)
College or above	−0.07 (1.72)	0.04 (1.39)	0.05 (1.40)	0.04 (1.30)	0.12 (4.10)	0.08 (2.47)	0.07 (1.95)
*p*	0.014	0.095	0.026	0.090	0.068	0.071	0.051
Household income per year, CNY							
<50,000	0.06 (1.74)	−0.08 (1.26)	−0.10 (1.12)	−0.03 (1.25)	−0.11 (3.60)	−0.09 (2.44)	−0.12 (1.96)
≥50,000	−0.07 (1.70)	0.09 (1.43)	0.11 (1.54)	0.04 (1.30)	0.12 (4.04)	0.11 (2.53)	0.13 (2.00)
*p*	0.244	0.042	0.010	0.379	0.324	0.186	0.037
Parity							
0	0.01 (1.71)	0.04 (1.37)	0.04 (1.40)	0.03 (1.28)	−0.03 (3.71)	0.12 (2.52)	0.04 (1.95)
≥1	−0.05 (1.75)	−0.13 (1.27)	−0.12 (1.08)	−0.11 (1.24)	0.12 (4.17)	−0.42 (2.29)	−0.16 (2.08)
*p*	0.618	0.095	0.103	0.130	0.580	0.003	0.164
Smoking status during pregnancy							
Yes	−0.95 (1.61)	−0.57 (0.77)	0.11 (1.78)	−0.33 (0.97)	0.03 (5.25)	−0.87 (2.82)	−1.50 (2.28)
No	0.01 (1.72)	0.00 (1.35)	0.00 (1.33)	0.00 (1.27)	0.00 (3.80)	0.01 (2.48)	0.01 (1.98)
*p*	0.145	0.264	0.828	0.491	0.978	0.349	0.044
Pre-pregnancy BMI category, kg/m^2^							
<18.5	−0.09 (1.74)	−0.18 (1.23)	0.00 (1.25)	0.04 (1.33)	0.18 (4.23)	0.18 (2.91)	0.35 (2.03)
18.5–<24.0	0.05 (1.74)	−0.03 (1.31)	−0.04 (1.20)	−0.03 (1.27)	0.00 (3.82)	0.02 (2.43)	0.07 (2.01)
≥24.0	−0.07 (1.66)	0.15 (1.47)	0.09 (1.63)	0.06 (1.25)	−0.08 (3.59)	−0.13 (2.38)	−0.32 (1.85)
*p*	0.484	0.049	0.413	0.553	0.801	0.457	0.002
Physical Activity, MET-hour/week							
<100	−0.04 (1.69)	0.00 (1.51)	0.09 (1.71)	−0.01 (1.42)	−0.18 (2.92)	−0.22 (2.17)	−0.29 (1.95)
100 to <200	0.07 (1.72)	0.00 (1.28)	−0.02 (1.14)	0.03 (1.22)	−0.11 (3.73)	−0.01 (2.43)	0.08 (1.87)
≥200	−0.12 (1.75)	0.00 (1.32)	−0.06 (1.26)	−0.08 (1.22)	0.54 (4.88)	0.31 (2.94)	0.15 (2.26)
*p*	0.343	1.000	0.415	0.529	0.066	0.059	0.015
Energy intake, kcal/d							
<2100	0.70 (1.24)	−0.05 (1.15)	−0.16 (1.14)	−0.22 (1.10)	−0.03 (3.28)	0.02 (2.53)	−0.03 (2.02)
≥2100	1.17 (1.77)	0.09 (1.63)	0.26 (1.58)	0.36 (1.45)	0.05 (4.57)	−0.03 (2.40)	0.05 (1.91)
*p*	<0.001	0.120	<0.001	<0.001	0.730	0.785	0.548
History of hypertension							
Yes	1.29 (0.45)	0.46 (3.72)	−0.70 (0.92)	−2.04 (0.82)	2.04 (1.07)	2.58 (3.39)	0.15 (2.80)
No	0.00 (1.72)	0.00 (1.34)	0.00 (1.33)	0.01 (1.27)	−0.01 (3.81)	−0.01 (2.47)	0.00 (1.98)
*p*	0.192	0.556	0.361	0.006	0.353	0.071	0.897
History of diabetes mellitus							
Yes	0.00 (1.72)	0.00 (1.34)	0.00 (1.33)	0.00 (1.27)	0.00 (3.81)	0.00 (2.48)	0.00 (1.98)
No	0.96 (0.99)	−1.14 (3.19)	−0.69 (0.14)	−1.37 (2.16)	−0.17 (2.98)	0.08 (2.21)	−0.54 (0.76)
*p*	0.429	0.232	0.463	0.128	0.950	0.963	0.698
Gestational diabetes mellitus							
Yes	0.10 (1.78)	−0.03 (1.32)	0.05 (1.42)	0.04 (1.30)	0.09 (4.23)	0.08 (2.54)	0.14 (1.93)
No	−0.22 (1.55)	0.29 (1.42)	−0.12 (1.09)	−0.03 (1.22)	−0.14 (2.56)	−0.09 (2.27)	−0.42 (2.01)
*p*	0.017	0.003	0.098	0.456	0.453	0.357	<0.001

SD: standard deviation, TFD: three-day food diary, FFQ: food frequency questionnaire, CNY: Chinese Yuan, BMI: body mass index, MET: metabolic equivalent.

**Table 3 nutrients-14-04342-t003:** Associations of maternal dietary pattern scores during pregnancy, in quartiles, with mean blood pressure throughout pregnancy.

Dietary Patterns	Diastolic Blood Pressure					Systolic Blood Pressure					**Mean Arterial Pressure**				
	Q1 Ref.	Q2 β (95%CI)	Q3 β (95%CI)	Q4 β (95%CI)	*p* _for Trend_	Q1 Ref.	Q2 β (95%CI)	Q3 β (95%CI)	Q4 β (95%CI)	*p* _for Trend_	**Q1** **Ref.**	**Q2 β** **(95%CI)**	**Q3 β** **(95%CI)**	**Q4 β** **(95%CI)**	** *p* ** ** _for Trend_ **
**Three-Day Food Diaries**															
Traditional (TFD)															
Model 1	0.00	0.09 (−1.07, 1.07)	0.10 (−0.98, 1.18)	0.41 (−0.67, 1.49)	0.417	0.00	−0.09 (−1.76, 1.58)	0.65 (−1.03, 2.32)	1.17 (−0.51, 2.84)	0.111	0.00	−0.07 (−1.27, 1.12)	0.27 (−0.93, 1.47)	0.62 (−0.57, 1.82)	0.235
Model 2	0.00	0.12 (−0.97, 1.20)	0.17 (−0.92, 1.26)	0.53 (−0.57, 1.62)	0.359	0.00	0.02 (−1.66, 1.69)	0.55 (−1.13, 2.23)	1.00 (−0.69, 2.70)	0.131	0.00	0.03 (−1.18, 1.23)	0.27 (−0.94, 1.48)	0.64 (−0.58, 1.86)	0.233
Model 3	0.00	0.05 (−1.01, 1.11)	0.21 (−0.94, 1.35)	0.71 (−0.61, 2.02)	0.249	0.00	−0.25 (−1.87, 1.36)	0.34 (−1.41, 2.08)	0.93 (−1.06, 2.93)	0.218	0.00	−0.11 (−1.28, 1.05)	0.21 (−1.04, 1.47)	0.71 (−0.72, 2.15)	0.232
Wheaten food-coarse cereals (TFD)															
Model 1	0.00	0.35 (−0.72, 1.41)	0.44 (−0.65, 1.54)	0.01 (−0.98, 1.17)	0.856	0.00	−0.81 (−2.46, 0.84)	−0.95 (−2.65, 0.74)	−0.97 (−2.65, −0.70)	0.261	0.00	−0.01 (−1.19, 1.18)	−0.04 (−1.26, 1.17)	−0.25 (−1.45, 0.95)	0.677
Model 2	0.00	0.35 (−0.73, 1.42)	0.47 (−0.64, 1.58)	0.15 (−0.95, 1.25)	0.688	0.00	−0.83 (−2.50, 0.83)	−0.81 (−2.52, 0.91)	−0.87 (−2.57, 0.83)	0.351	0.00	−0.02 (−1.22, 1.18)	0.02 (−1.22, 1.25)	−0.20 (−1.42, 1.03)	0.820
Model 3	0.00	0.37 (−0.65, 1.40)	0.28 (−0.78, 1.34)	−0.12 (−1.18, 0.93)	0.847	0.00	−0.74 (−2.30, 0.82)	−1.21 (−2.82, 0.41)	−1.44 (−3.05, 0.17)	0.061	0.00	0.03 (−1.09, 1.15)	−0.25 (−1.41, 0.91)	−0.57 (−1.73, 0.59)	0.306
Sweet food-seafood (TFD)															
Model 1	0.00	0.05 (−1.02, 1.13)	0.31 (−0.76, 1.39)	−1.01 (−2.09, 0.06)	0.041	0.00	1.17 (−2.84, 0.49)	0.36 (−1.31, 2.03)	−2.31 (−3.98, −0.65)	0.014	0.00	−0.35 (−1.54, 0.85)	0.34 (−0.86, 1.53)	−1.41 (−2.60, −0.21)	0.022
Model 2	0.00	0.02 (−1.07, 1.10)	0.25 (−0.85, 1.35)	−1.04 (−2.15, 0.06)	0.028	0.00	−1.20 (−2.87, 0.48)	0.25 (−1.45, 1.95)	−2.47 (−4.18, −0.77)	0.007	0.00	−0.38 (−1.59, 0.82)	0.24 (−0.98, 1.46)	−1.50 (−2.72, −0.27)	0.011
Model 3	0.00	−0.09 (−1.13, 0.95)	0.04 (−1.01, 1.09)	−1.04 (−2.10, 0.02)	0.031	0.00	−1.53 (−3.11, 0.05)	−0.17 (−1.77, 1.43)	−2.57 (−4.19, −0.96)	0.005	0.00	−0.57 (−1.70, 0.57)	−0.05 (−1.20, 1.10)	−1.54 (−2.70, −0.38)	0.009
Fried food-protein-rich (TFD)															
Model 1	0.00	−0.03 (−1.11, 1.05)	−0.45 (−1.53, 0.63)	0.24 (−0.84, 1.31)	0.730	0.00	−1.06 (−2.73, 0.62)	−0.60 (−2.28, 1.07)	0.85 (−0.82, 2.52)	0.163	0.00	−0.33 (−1.53, 0.87)	−0.38 (−1.58, 0.82)	0.55 (−0.65, 1.75)	0.292
Model 2	0.00	0.33 (−0.75, 1.40)	−0.03 (−1.12, 1.06)	0.45 (−0.65, 1.55)	0.553	0.00	−0.92 (−2.60, 0.76)	−0.31 (−2.01, 1.39)	1.09 (−0.62, 2.79)	0.068	0.00	−0.28 (−1.49, 0.92)	−0.18 (−1.40, 1.04)	0.66 (−0.57, 1.89)	0.162
Model 3	0.00	0.12 (−0.92, 1.16)	−0.12 (−1.18, 0.94)	0.22 (−0.87, 1.31)	0.665	0.00	−0.73 (−2.32, 0.85)	0.05 (−1.67, 1.57)	0.81 (−0.85, 2.47)	0.141	0.00	−0.13 (−1.26, 1.01)	0.01 (−1.15, 1.18)	0.51 (−0.68, 1.71)	0.265
**Food Frequency Questionnaire**															
Fish-seafood (FFQ)															
Model 1	0.00	−1.00 (−2.08, 0.07)	0.20 (−0.87, 1.28)	0.49 (−0.58, 1.57)	0.083	0.00	−1.53 (−3.21, 0.13)	1.06 (−0.61, 2.74)	0.13 (−1.54, 1.80)	0.321	0.00	−1.06 (−2.25, 0.14)	0.53 (−0.67, 1.72)	0.47 (−0.72, 1.67)	0.108
Model 2	0.00	−1.07 (−2.23, 0.07)	0.13 (−1.07, 1.33)	0.48 (−0.82, 1.78)	0.101	0.00	−1.81 (−3.59, −0.03)	0.61 (−1.25, 2.47)	−0.53 (−2.55, 1.48)	0.801	0.00	−1.22 (−2.50, 0.06)	0.29 (−1.04, 1.63)	0.19 (−1.25, 1.64)	0.267
Model 3	0.00	−1.23 (−2.33, 0.14)	0.07 (−1.07, 1.22)	0.52 (−0.72, 1.77)	0.057	0.00	−2.17 (−3.84, −0.50)	0.39 (−1.35, 2.14)	−0.60 (−2.49, 1.30)	0.693	0.00	−1.44 (−2.64, −0.24)	0.18 (−1.07, 1.43)	0.20 (−1.16, 1.56)	0.176
Protein-rich (FFQ)															
Model 1	0.00	0.27 (−0.81, 1.35)	0.17 (−0.91, 1.25)	0.15 (−0.93, 1.22)	0.853	0.00	−0.58 (−2.26, 1.09)	0.64 (−1.04, 2.31)	0.52 (−1.16, 2.19)	0.314	0.00	0.00 (−1.20, 1.20)	0.39 (−0.81, 1.59)	0.36 (−0.83, 1.56)	0.453
Model 2	0.00	0.48 (−0.66, 1.62)	0.42 (−0.82, 1.66)	0.19 (−1.19, 1.57)	0.936	0.00	−0.24 (−2.01, 1.53)	1.15 (−0.76, 3.07)	1.07 (−1.07, 3.21)	0.204	0.00	0.26 (−1.01, 1.52)	0.74 (−0.64, 2.11)	0.59 (−0.95, 2.12)	0.486
Model 3	0.00	0.55 (−0.54, 1.64)	0.63 (−0.54, 1.81)	0.17 (−1.14, 1.48)	0.869	0.00	−0.18 (−1.84, 1.48)	1.47 (−0.33, 3.26)	1.03 (−0.97, 3.03)	0.214	0.00	0.32 (−0.88, 1.51)	0.97 (−0.32, 2.26)	0.56 (−0.88, 1.99)	0.522
Vegetable-fruit-rice (FFQ)															
Model 1	0.00	0.20 (−0.88, 1.27)	−0.73 (−1.81, 0.34)	−0.86 (−1.94, 0.22)	0.046	0.00	1.05 (−0.63, 2.72)	−0.97 (−2.64, 0.70)	−0.78 (−2.45, 0.89)	0.119	0.00	0.35 (−0.85, 1.55)	−0.85 (−2.05, 0.34)	−0.87 (−2.06, 0.33)	0.054
Model 2	0.00	0.16 (−0.94, 1.26)	−0.74 (−1.86, 0.38)	−0.96 (−2.12, 0.21)	0.051	0.00	0.89 (−0.81, 2.60)	−1.21 (−2.95, 0.52)	−1.22 (−3.02, 0.59)	0.046	0.00	0.25 (−0.97, 1.47)	−0.97 (−2.22, 0.27)	−1.12 (−2.41, 0.17)	0.030
Model 3	0.00	0.05 (−1.00, 1.11)	−0.45 (−1.52, 0.62)	−0.48 (−1.60, 0.63)	0.331	0.00	0.75 (−0.85, 2.36)	−0.78 (−2.42, 0.86)	−0.41 (−2.11, 1.29)	0.345	0.00	0.12 (−1.03, 1.28)	−0.65 (−1.82, 0.53)	−0.55 (−1.77, 0.67)	0.259

Model 1: Crude model; Model 2: Adjusted for other dietary pattern scores from the same dietary assessment tool (TFD or FFQ); Model 3: Model 2 + age, parity, family income, education level, ethnicity, smoking status, total energy intake per day, physical activity status per week, pre-pregnancy BMI, history of hypertension and history of diabetes. Q1: quartile 1, Q2: quartile 2, Q3: quartile 3, Q4: quartile 4, CI: confidence interval, TFD: three-day food dairies, FFQ: food frequency questionnaire, BMI: body mass index.

**Table 4 nutrients-14-04342-t004:** Associations of maternal dietary pattern scores during pregnancy, in quartiles, with risk of gestational hypertension.

Dietary Patterns	Gestational Hypertension (*n* = 1082)				
	Q1 Reference	Q2 OR (95%CI)	Q3 OR (95%CI)	Q4 OR (95%CI)	*p* _for Trend_
**Three-Day Food Diaries**					
Traditional (TFD)					
Model 1	1.00	0.83 (0.52, 1.34)	0.68 (0.42, 1.12)	0.90 (0.56, 1.45)	0.661
Model 2	1.00	0.81 (0.50, 1.32)	0.62 (0.38, 1.03)	0.80 (0.49, 1.29)	0.492
Model 3	1.00	0.79 (0.48, 1.29)	0.60 (0.35, 1.03)	0.74 (0.42, 1.31)	0.347
Wheaten food-coarse cereals (TFD)					
Model 1	1.00	0.73 (0.46, 1.17)	0.60 (0.36, 1.00)	0.82 (0.52, 1.31)	0.341
Model 2	1.00	0.68 (0.42, 1.10)	0.56 (0.33, 0.94)	0.76 (0.47, 1.23)	0.223
Model 3	1.00	0.68 (0.41, 1.10)	0.53 (0.31, 0.90)	0.71 (0.43, 1.16)	0.122
Sweet food-seafood (TFD)					
Model 1	1.00	0.80 (0.49, 1.31)	1.02 (0.63, 1.63)	0.86 (0.53, 1.40)	0.737
Model 2	1.00	0.80 (0.48, 1.32)	1.00 (0.62, 1.64)	0.84 (0.51, 1.64)	0.579
Model 3	1.00	0.79 (0.48, 1.33)	0.93 (0.57, 1.54)	0.80 (0.48, 1.35)	0.474
Fried food-protein-rich (TFD)					
Model 1	1.00	0.79 (0.47, 1.33)	0.92 (0.55, 1.51)	1.43 (0.90, 2.28)	0.049
Model 2	1.00	0.82 (0.48, 1.38)	0.94 (0.56, 1.58)	1.55 (0.96, 2.51)	0.027
Model 3	1.00	0.86 (0.51, 1.46)	0.98 (0.58, 1.65)	1.51 (0.91, 2.51)	0.052
**Food Frequency** **Questionnaire**					
Fish-seafood (FFQ)					
Model 1	1.00	0.81 (0.49, 1.35)	1.13 (0.70, 1.83)	1.14 (0.70, 1.84)	0.349
Model 2	1.00	0.77 (0.44, 1.32)	1.05 (0.61, 1.80)	1.03 (0.58, 1.86)	0.597
Model 3	1.00	0.74 (0.42, 1.29)	1.07 (0.62, 1.85)	1.04 (0.57, 1.88)	0.567
Protein-rich (FFQ)					
Model 1	1.00	0.87 (0.52, 1.44)	1.23 (0.76, 1.98)	1.10 (0.67, 1.79)	0.451
Model 2	1.00	0.91 (0.53, 1.56)	1.29 (0.74, 2.27)	1.13 (0.60, 2.14)	0.586
Model 3	1.00	0.89 (0.52, 1.55)	1.35 (0.76, 2.39)	1.14 (0.60, 2.17)	0.552
Vegetable-fruit-rice (FFQ)					
Model 1	1.00	0.99 (0.61, 1.60)	1.01 (0.62, 1.63)	0.88 (0.54, 1.45)	0.651
Model 2	1.00	0.94 (0.57, 1.56)	0.95 (0.57, 1.58)	0.81 (0.47, 1.39)	0.488
Model 3	1.00	0.89 (0.53, 1.48)	0.93 (0.55, 1.57)	0.84 (0.48, 1.46)	0.643

Model 1: Crude model; Model 2: Adjusted for other dietary pattern scores from the same dietary assessment tool (TFD or FFQ); Model 3: Model 2 + age, parity, family income, education level, ethnicity, smoking status, total energy intake per day, physical activity status per week, pre-pregnancy BMI, and history of diabetes; Q1: quartile, Q2: quartile 2, Q3: quartile 3, Q4: quartile 4, CI: confidence interval, TFD: three-day food dairies, FFQ: food frequency questionnaire, BMI: body mass index.

## Data Availability

Data presented in this study are available on request from thecorresponding author.

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
