# Peer review of "Associations of Dietary Patterns during Pregnancy with Gestational Hypertension: The “Born in Shenyang” Cohort Study"

_nutrients, 2022, doi:10.3390/nu14204342_

Round 1
Reviewer 1 Report
Reviewer report
Pre-pregnancy weight status and associations of dietary pat-2 terns with maternal blood pressure during pregnancy: the 3 “Born in Shenyang” cohort
- Line 89-90….” we enrolled pregnant women with single pregnancies at 54 com-89 munity health care centers and hospitals in Shenyang, China, from April to September 90 2017. Participants were recruited between 21 and 24 weeks of gestation". Why you recruited between 21 and 24 weeks of gestation?
- Line 109….” We assessed maternal diet during pregnancy using both three-day food diaries (TFD) 109 and food frequency questionnaires (FFQ)”., so if you assessed the dietary pattern during pregnancy, the title better be changed in such a way.
- Line 109-128….was the validity and reliability of dietary assessment tools (TFD and FFQ) measured /assessed and how?
- Line 121…The FFQ is usually used to measure food intake over a specified period, but the following statement seems that FFQ measures one-day food consumption. “The FFQ contains 25 food items with nine consumption frequency categories 121 ranging from “never” to “more or equal to 3 times per day”
- As the authors measured the dietary intake of pregnant women using two measures (TFD and FFQ), which data was used to determine the association between dietary pattern and gestational hypertension?
- Line 142-157. although pre-pregnancy BMI is the primary exposure variable of this study, nothing stated how pre-pregnancy BMI was measured. Even it is not clear when and how the participants' weight was measured.
- The study did not incorporate potential confounding variables like previous pregnancy history of gestational hypertension and family history of hypertension.
- The analysis did not address the pre-pregnancy BMI
- The result part is not concise and presented clearly; lots of descriptions repeat the content already presented in the tables.
Reviewer 2 Report
Hu et al report the association between maternal eating patterns and blood pressure in the already described “Born in Shenyang Cohort”.
General comments:
The authors report that almost all the eating patterns identified are associated with either reduced systolic or diastolic pressure or with reduced risk of gestational hypertension. The conclusions to be drawn from the results are not clear, statistical significance is not reached by most of the comparisons and the authors do not give concrete dietary recommendations.
There is a high degree of overlap between the submitted manuscript and references 18 and 19. Indeed, as an example, table 2 has been published in reference 18 already.
It is not clear to this reviewer what this work adds to preexisting knowledge.
A revision by a native English or professional editor is advisable
Specific comments:
Introduction: Prevention of gestational hypertension is suggested by several dietary interventions, but the references given are observational studies, not clinical trials. An association, but not causality, can be claimed.
Methods: pre-pregnancy weight is self-reported. The authors should consider not including this variable as a central aspect of the work, and omitting it altogether from the title
Only gestational hypertension is considered as an outcome, but not pre-eclampsia, which is what is really associated with maternal and foetal outcomes.
Diabetes is associated with increased risk of gestational hypertension and pre-eclampsia, but is not mentioned at all in the paper.
Results: most of the results claimed are not statistically significant. The implications of the results are not clear
Reviewer 3 Report
This manuscript is written well. However, it will be able to revise a few points.
1) Authors mentioned that to address these gaps, we examined the association of maternal pre-pregnancy BMI //// in the Introduction. Could you replace this sentence after hypothesis?
2) Are there any other study limitations here?
Round 2
Reviewer 1 Report
The authors have addressed most concerns raised.